# Ajuforrestin A Inhibits Tumor Proliferation and Migration by Targeting the STAT3/FAK Signaling Pathways and VEGFR-2

**DOI:** 10.3390/biology14080908

**Published:** 2025-07-22

**Authors:** Sibei Wang, Yeling Li, Mingming Rong, Yuejun Li, Yaxin Lu, Shen Li, Dongho Lee, Jing Xu, Yuanqiang Guo

**Affiliations:** 1State Key Laboratory of Medicinal Chemical Biology, College of Pharmacy, and Tianjin Key Laboratory of Molecular Drug Research, Nankai University, Tianjin 300350, China; sibeiwang2017@163.com (S.W.); liyeling320@163.com (Y.L.); 18519232516@163.com (M.R.);; 2College of Chemistry, Nankai University, Tianjin 300071, China; 3Department of Plant Biotechnology, College of Life Sciences and Biotechnology, Korea University, Seoul 02841, Republic of Korea; dongholee@korea.ac.kr

**Keywords:** natural diterpenoid, antitumor activity, STAT3, FAK, VEFGR-2, zebrafish

## Abstract

Representing 85% of all lung malignancies, non-small cell lung cancer (NSCLC) presents significant therapeutic challenges. This research explored the anticancer properties of ajuforrestin A, a natural compound from *Ajuga lupulina* Maxim, against A549 lung carcinoma cells. Treatment with ajuforrestin A potently suppressed the viability of A549 cells, yielding an IC_50_ value of 9.0 μM. The compound was found to trigger apoptosis, elevating the apoptotic cell population to 40.5% at 20 μM from a baseline of 10.4% in untreated controls. It also prompted a G0/G1 phase cell cycle blockade, increasing the proportion of cells in the G1 phase to 88.1% at 20 μM, compared to 70.7% for the control. Mechanistically, ajuforrestin A governed the expression of associated proteins by regulating the STAT3 pathway and curtailed cellular migration via suppression of the FAK cascade. Zebrafish-based assays confirmed that ajuforrestin A obstructed angiogenesis. Furthermore, surface plasmon resonance (SPR) analysis demonstrated its strong binding affinity for VEGFR-2, with a dissociation constant (KD) of 45 μM. In a living model, ajuforrestin A at 20 μM achieved inhibition rates of 80.0% for A549 cell proliferation and 66.1% for metastasis. These findings underscore the significant in vitro and in vivo antitumor potential of ajuforrestin A, offering a novel perspective for NSCLC therapy.

## 1. Introduction

Lung cancer represents a paramount global health challenge, with one of the highest mortality rates among malignant neoplasms. Among individuals aged over 50, it is diagnosed in approximately one in five cancer patients, and its incidence continues to rise annually [1]. Histologically, lung carcinomas are categorized into small cell lung cancer (SCLC) and non-small cell lung cancer (NSCLC), with NSCLC constituting the vast majority (approximately 85%) of cases [2]. The absence of distinct clinical symptoms during early disease stages and the constraints of current screening methods mean that a majority of individuals are diagnosed at an advanced stage. Conventional therapeutic modalities, including surgery, chemotherapy, radiotherapy, and targeted treatments, are frequently hampered by late detection, the emergence of drug resistance, and significant adverse effects, leading to a generally poor prognosis and low survival rates [3]. Consequently, there is an imperative to identify novel therapeutic targets for lung cancer and to pursue the discovery and development of molecularly targeted agents that possess high efficacy and minimal toxicity for treating this and other malignancies.

Apoptosis, or programmed cell death, constitutes a meticulously regulated cellular self-destruction process characterized by distinct morphological and biochemical features [4]. In physiological contexts, the body utilizes apoptosis to eliminate senescent or aberrant cells, thereby preserving tissue homeostasis. In the context of cancer, the inhibition of this process leads to uncontrolled malignant cell proliferation and subsequent carcinogenesis. The mechanisms of apoptosis are therefore intimately linked with the onset, progression, and therapeutic response of tumors. Leveraging the induction and regulatory machinery of apoptosis to eliminate tumor cells has emerged as a cornerstone of modern cancer therapy.

Signal transducers and activators of transcription (STAT) proteins comprise a family of transcription factors that participate in numerous pathways triggered by both extrinsic and intrinsic cues, culminating in the expression or repression of target genes governing cell proliferation, apoptosis, inflammation, and angiogenesis [5]. Research has shown that STAT3, a key member of this family, is frequently overexpressed and constitutively active in various tumor cells and tissues. While STAT3 itself may not inherently possess oncogenic or tumor-suppressive functions, its activation through phosphorylation leads to dimerization and nuclear translocation. Once in the nucleus, it modulates the expression of downstream proteins, thereby influencing critical cellular functions and promoting cancer development and progression.

Furthermore, cellular migration is a prerequisite for tumor cell invasion and the establishment of metastases. Focal adhesion kinase (FAK), a non-receptor tyrosine kinase encoded by the Protein Tyrosine Kinase 2 (PTK2) gene, is a central regulator of this process. FAK participates in cell adhesion dynamics and cytoskeletal rearrangement, and it also contributes to the regulation of apoptosis [6]. Aberrant upregulation and hyperactivation of FAK are observed in a wide array of human malignancies, including lung cancer, gastric cancer, and melanoma. Overexpression of FAK facilitates the dislodgement of malignant cells from their primary locus, allowing them to circulate via the blood and lymphatic systems to distant sites, such as the lungs and liver, where they can form new metastatic colonies. As a critical intracellular signaling hub, FAK therefore plays a central role in tumor invasion and metastasis. Additionally, tumor growth and metastasis are heavily reliant on neovascularization [7]. The VEGF/VEGFR-2 signaling axis is known to orchestrate the migration, proliferation, and survival of neoplastic endothelial cells, making it a pivotal pathway in angiogenesis. Consequently, targeting the VEGF/VEGFR-2 pathway presents a viable strategy for cancer treatment.

Natural products offer a rich chemical diversity, a wide spectrum of biological activities, and often reduced side effects, making them an invaluable resource for cancer therapy [8]. A multitude of small molecules sourced from nature have demonstrated anti-lung cancer activity by modulating specific genes and signaling pathways implicated in tumorigenesis [9]. *Ajuga lupulina* Maxim, a plant from the Labiatae family found in the Sichuan, Yunnan, and Qinghai provinces of China, is used in folk medicine for various ailments [10]. This plant is a rich source of diterpenoids and flavonoids, which exhibit a range of biological properties, including anticancer, antibacterial, and anti-inflammatory activities. In our prior work, we successfully isolated ajuforrestin A, an abietane-type diterpenoid, from this plant for the first time and noted its potent inhibitory effect on A549 cell proliferation. Given the high mortality associated with lung cancer, investigating natural compounds and their mechanisms of action is a primary focus of anticancer drug discovery [9].

In our pursuit of novel agents against lung cancer, extensive screening identified ajuforrestin A as having remarkable antiproliferative activity against A549 cells. This study was therefore undertaken to elucidate its antitumor mechanisms and evaluate its therapeutic potential. We explored the impact of ajuforrestin A on key cancer hallmarks, namely unlimited proliferation and migration. Recognizing that tumor angiogenesis is a critical component of cancer progression, we also investigated the compound’s effect on VEGFR-2 and associated neovascularization. Finally, we assessed the in vivo anti-lung cancer efficacy of ajuforrestin A using a zebrafish xenograft tumor model.

## 2. Materials and Methods

Ajuforrestin A, a natural diterpenoid, was successfully isolated and purified from *A. lupulina* Maxim. The compound’s ^1^H and ^13^C NMR spectra are provided in the Appendix A.

### 2.1. Reagents and Materials

Details regarding commercial materials, chemical reagents, their manufacturers, and related information are provided in the Appendix A.

### 2.2. Assessment of Cell Viability In Vitro

The cytotoxic effect of ajuforrestin A on the A549 cell line, obtained from the Cell Bank of the Chinese Academy of Sciences (Shanghai, China), was quantified using the MTT method. A detailed description of the experimental protocol can be found in the Appendix A.

### 2.3. Flow Cytometric Analysis of Apoptosis

Following the manufacturer’s protocol for the Annexin V-FITC Apoptosis Detection Kit, the influence of ajuforrestin A on A549 cell apoptosis was evaluated via flow cytometry. The specific experimental steps are detailed in the Appendix A.

### 2.4. Cell Cycle Distribution Analysis

Flow cytometry was employed to ascertain the effects of ajuforrestin A on the cell cycle progression of A549 cells [11,12]. A full description of the cell cycle experiment is contained within the Appendix A.

### 2.5. Cell Migration Analysis via Wound-Healing Assay

The impact of ajuforrestin A on the migratory capacity of A549 cells was evaluated using a wound-healing (scratch) assay [13,14]. The procedural details are accessible in the Appendix A.

### 2.6. Western Blotting Analysis

The expression levels of pertinent proteins were quantitatively determined through Western blotting [15]. Comprehensive protocols for this analysis are furnished in the Appendix A.

### 2.7. Zebrafish Husbandry and Care

Adult AB strain zebrafish were procured from Shanghai Feixi Biotechnology Co., Ltd. (Shanghai, China). The procedures for zebrafish maintenance and breeding, following established methods [16], are included in the Appendix A.

### 2.8. In Vivo Anti-Angiogenetic Activity of Ajuforrestin A in Zebrafish

The anti-angiogenic potential of ajuforrestin A was assessed within a transgenic *Tg*(*fli1:EGFP*) zebrafish model. The zebrafish were procured from Shanghai Feixi Biotechnology Co., Ltd. (Shanghai, China). The methodology for this experiment is described in the Appendix A.

### 2.9. Surface Plasmon Resonance (SPR) Assay

The binding affinity between ajuforrestin A and the VEGFR-2 protein was measured by SPR analysis. The detailed protocol for this assay is outlined in the Appendix A.

### 2.10. Molecular Docking Simulation

To further explore the interaction mechanism between ajuforrestin A and VEGFR-2, AutoDock Vina software (version 1.1.2) was used for molecular docking simulation [17]. The VEGFR-2 protein structure (PDB ID: 4ASE) was obtained from the PDB database [18]. Firstly, ajuforrestin A and the VEGFR-2 receptor protein were preprocessed. The center coordinates of the grid box were (−23.66, 0.77, −12.66), the size of the grid box was set as 40 × 40 × 40 Å, and the grid spacing was 0.375 Å. The Lamarckian genetic algorithm was used to combine the local energy search with the genetic algorithm, and the semiempirical potential function was used as the energy scoring function to perform the global search for the small molecule conformation and position, while other parameters were set by default. Finally, the structures of the docked complexes were analyzed by PyMOL (version 3.0.5) and LigPlot software (version 2.2.9) [19,20].

### 2.11. In Vivo Antitumor Activity of Ajuforrestin A in Zebrafish

In vivo antitumor effects of ajuforrestin A on A549 cells were evaluated by establishing a zebrafish tumor xenotransplantation model. The specific methods used are available in the Appendix A.

### 2.12. Statistical Analysis

All statistical analyses were conducted with GraphPad Prism 6.0 (GraphPad Software, San Diego, CA, USA). Data are presented as the mean ± standard deviation (SD) from three independent experiments (*n* = 3). Differences between groups were assessed using one-way analysis of variance (ANOVA) followed by Tukey’s post hoc test. A *p*-value less than 0.05 was deemed statistically significant.

## 3. Results

### 3.1. In Vitro Antiproliferative Activity

Ajuforrestin A, an abietane-type diterpenoid previously isolated by our team from *A. lupulina* Maxim, was evaluated for its antiproliferative effects. The MTT assay determined that ajuforrestin A markedly suppressed the growth of A549 cells in vitro, with a calculated half-maximal inhibitory concentration (IC_50_) of 9.0 μM (Table 1). Based on this potent activity, we proceeded to investigate its potential mechanism of action and the associated signaling pathways.

To ascertain the compound’s safety profile, its cytotoxicity against non-cancerous cells was measured. The selectivity index (SI), calculated as the ratio of the IC_50_ for a normal cell line (HEK29T) to that of the cancer cell line, provides a measure of anticancer specificity. Compounds with an SI greater than 3 are typically considered highly selective [21]. The MTT assay revealed that ajuforrestin A had low cytotoxicity toward the non-malignant HEK293T cell line, with an IC_50_ of 126.89 ± 3.523 μM. This resulted in an SI value of 14.10, which indicates excellent safety for normal cells and exceptionally high selectivity toward A549 cells.

### 3.2. Ajuforrestin A Triggered Apoptosis in A549 Cells

Apoptosis is a primary mode of regulated cell death. To determine if ajuforrestin A inhibits A549 cell proliferation by inducing apoptosis, we utilized flow cytometry. Following treatment with increasing doses of ajuforrestin A (5, 10, and 20 μM), apoptosis was quantified via Annexin V-FITC and PI co-staining. As illustrated in Figure 1, the fraction of apoptotic cells rose from 10.4% in the control to 13.1% (5 μM), 23.6% (10 μM), and reached 40.5% (20 μM). These observations confirm that ajuforrestin A can curtail cell proliferation by stimulating apoptosis in A549 cells.

### 3.3. Ajuforrestin A Induced G0/G1 Cell Cycle Arrest in A549 Cells

Given that cell cycle arrest is a significant contributor to apoptosis [22], we investigated the influence of ajuforrestin A on the cell cycle. After drug exposure and PI staining, cell cycle distribution was assessed by flow cytometry. As shown in Figure 2, as the ajuforrestin A concentration was raised from 0 to 20 μM, the percentage of cells in the G1 phase increased from 70.7% (control) to 76.7% (5 μM), 85.8% (10 μM), and 88.1% (20 μM). This evidence demonstrates that ajuforrestin A can arrest A549 cells at the G0/G1 transition.

### 3.4. Ajuforrestin A Modulated the STAT3-Associated Signaling Pathway

STAT3 is a well-documented oncogenic factor that is highly expressed across numerous tumor types [23]. Extensive research confirms that STAT3 is aberrantly activated in many cancer cells, linking it directly to tumorigenesis, with tumor growth being regulated by STAT3 and its downstream targets. To clarify the mechanism of ajuforrestin A in A549 cells, the expression of STAT3 and its related proteins was examined using Western blotting. The results revealed that while ajuforrestin A did not alter the total expression of STAT3 protein, it significantly and dose-dependently suppressed the phosphorylation of STAT3 at the Tyr705 residue (Figure 3).

Because STAT3 activation regulates the expression of proteins involved in cell survival, proliferation, and angiogenesis, we further explored the effect of ajuforrestin A on STAT3 downstream targets. As depicted in Figure 3, the expression of the anti-apoptotic protein Bcl-2 was diminished, while the expression of the pro-apoptotic protein Bax was augmented. Furthermore, members of the caspase family, such as caspase-3 and caspase-9, are critical executioners in the apoptotic cascade [4]. Our Western blotting data showed that the levels of cleaved caspase-3 and cleaved caspase-9 increased. These results collectively indicate that ajuforrestin A may exert its antiproliferative activity in A549 cells via regulation of the STAT3-mediated apoptotic pathway.

### 3.5. Ajuforrestin A Inhibited Cell Migration via Regulation of the FAK Signaling Pathway

Beyond proliferation, cell migration is another critical determinant in tumor progression [24]. We therefore investigated the inhibitory capacity of ajuforrestin A on cell migration using a wound-healing assay. As shown in Figure 4A,B, the migration rates of A549 cells at 48 h were 67.5% (control), 63.8% (7.5 μM), 31.7% (15 μM), and 19.1% (30 μM), demonstrating that ajuforrestin A could markedly impede the migration of A549 cells.

Tumor migration is governed by numerous proteins, including focal adhesion kinase (FAK) and its downstream effectors [6]. In cancer cells, overexpressed FAK contributes to focal adhesion formation and activates signaling cascades related to proliferation, migration, and angiogenesis. The phosphorylation of FAK at Tyr397 ultimately elevates the levels of matrix metalloproteinases (MMPs), which are crucial for tumor invasion. Consequently, the mechanism by which ajuforrestin A inhibits A549 cell migration was examined by Western blot. As illustrated in Figure 4C,D, total FAK protein expression remained unchanged, whereas phosphorylated FAK levels were substantially reduced in a dose-dependent manner following ajuforrestin A treatment. Matrix metalloproteinase 2 (MMP2), which facilitates tumor cell migration and invasion by disrupting the basement membrane and degrading the extracellular matrix, also showed reduced expression compared to the control. These findings suggest that ajuforrestin A inhibits tumor migration by targeting key proteins within the FAK signaling pathway.

### 3.6. In Vivo Anti-Angiogenic Efficacy of Ajuforrestin A

The formation of new vasculature is a vital component of the tumor microenvironment, supplying essential oxygen and nutrients for tumor growth and creating conduits for metastasis [25]. Tumor-associated angiogenesis is a key process throughout all stages of cancer development. The zebrafish model offers numerous advantages, including external fertilization, rapid development, optical transparency for real-time imaging, and high genomic homology with humans [26]. We therefore employed a transgenic *Tg*(*fli1:EGFP*) zebrafish model to observe the impact of ajuforrestin A on angiogenesis. As shown in Figure 5, control embryos developed normal, intact blood vessels. In contrast, zebrafish embryos treated with ajuforrestin A (5, 10, and 20 μM) or the positive control sunitinib (2 μM) exhibited damaged and fragmented intersegmental vessels (ISVs) and dorsal longitudinal anastomotic vessels (DLAVs). Quantitative analysis revealed that the average ISV length in the control group was 2998.6 ± 139.9 μm. Ajuforrestin A treatment led to a dose-dependent reduction in ISV length (2722.1 ± 132.4 μm at 5 μM, 2176.0 ± 151.6 μm at 10 μM, and 1904.7 ± 83.5 μm at 20 μM). These results demonstrate that ajuforrestin A can significantly inhibit angiogenesis in a living organism.

### 3.7. Ajuforrestin A Interacted with VEGFR-2 as Determined by SPR

Tumorigenesis results from the synergistic effects of multiple pro-cancerous factors. Tumor growth, invasion, and metastasis are all linked to angiogenesis, a process regulated by a balance of pro- and anti-angiogenic molecules. VEGF is the most prominent pro-angiogenic factor in this process [7]. It exerts its biological effects primarily by binding to VEGF receptors (VEGFRs) on the endothelial cell surface, with VEGFR-2 being the principal functional receptor [27]. The VEGF/VEGFR-2 signaling pathway is thus a critical driver of tumor angiogenesis, making its blockade a promising anticancer strategy.

SPR is a technique that provides real-time kinetic and thermodynamic data on intermolecular binding. It is widely used for screening drug candidates, identifying drug targets, and analyzing molecular interactions [28]. Our zebrafish experiments showed that ajuforrestin A could potently inhibit angiogenesis in vivo. To investigate whether this anti-angiogenic effect was mediated through VEGFR-2, we immobilized the protein on a sensor chip and monitored its interaction with ajuforrestin A. The results yielded a binding dissociation constant (KD) of 45 μM, indicating a strong affinity between ajuforrestin A and VEGFR-2 and suggesting that ajuforrestin A may exert its anti-angiogenic function by targeting this receptor (Figure 6).

### 3.8. Molecular Docking Simulation Insights

To further elucidate the binding mechanism of ajuforrestin A with VEGFR-2, we conducted molecular docking using Autodock Vina. The co-crystallized ligand of VEGFR-2 was successfully re-docked into its binding site, reproducing the original conformation with an RMSD of 1.027 Å (<2 Å), thereby validating the docking protocol. As shown in Figure 7, ajuforrestin A was predicted to form hydrogen bonds with residues Asp814, Lys868, and Asp1046. Additionally, it engaged in hydrophobic interactions with Asp814, Lys868, Glu885, Ile888, Leu889, Asp1046, Gly1048, and Leu1049. The calculated free binding energy of the ajuforrestin A–VEGFR-2 complex was –8.3 kcal/mol. These interactions are likely key to the inhibition of VEGFR-2 activity by ajuforrestin A. The VEGFR-2 protein is known to possess two crucial sub-pockets for small molecule binding: a solvent-exposed hydrophobic cleft (sub-site I) and a deeper, non-polar posterior pocket (sub-site II), accessible via Val916 [29]. The hinge region residues Cys919 and Glu917 form critical hydrophilic interactions essential for inhibitory activity, while Val914, Ala866, Cys1045, and Leu1035 constitute the hydrophobic pocket. Key interactions with Asp1046 and Glu885 are mediated by hydrogen bond acceptors and donors. Furthermore, Leu889, Ile892, Val898, and Ile1044 contribute to a hydrophobic or metastable pocket [30].

A comparative docking analysis with the known inhibitor sunitinib revealed that sunitinib formed a hydrogen bond with Glu885 and hydrophobic interactions with Asp1046, Lys868, and several other residues. Notably, ajuforrestin A shared six identical interacting amino acid residues with sunitinib under the same docking conditions (Figure 7C,D). This overlap suggests that ajuforrestin A and sunitinib may have similar VEGFR-2 inhibitory mechanisms. The presence of distinct interacting residues also hints that ajuforrestin A might overcome potential sunitinib resistance. These findings provide a theoretical foundation for the molecular design of novel VEGFR-2 inhibitors and aid the search for more effective antitumor agents.

### 3.9. In Vivo Antitumor Efficacy of Ajuforrestin A in Zebrafish Xenografts

To assess the inhibitory potential of ajuforrestin A on tumor growth and metastasis in vivo, we established a zebrafish xenograft model by injecting CM-DiI-labeled A549 cells into the yolk sac. The embryos were then treated with different concentrations of ajuforrestin A (5, 10, and 20 μM) or the positive control etoposide (10 μM). The antitumor effect was quantified by measuring the red fluorescence. As depicted in Figure 8, both the relative intensity and the number of disseminated foci of red fluorescence diminished in a dose-dependent manner with increasing ajuforrestin A concentration. At the highest dose of 20 μM, ajuforrestin A achieved inhibition rates of 80.0% for A549 cell proliferation and 66.1% for metastasis. These results affirm that ajuforrestin A can effectively suppress tumor proliferation and metastatic spread, demonstrating significant in vivo antitumor activity.

## 4. Discussion

Lung cancer is the primary cause of cancer-related mortality globally, responsible for approximately 18% of all cancer deaths and ranking as the second most diagnosed malignancy after breast cancer [31]. NSCLC comprises 85% of all lung cancers and carries the highest incidence and death rates. The severe adverse effects of conventional chemotherapy, high rates of acquired resistance to targeted therapies, and the immunosuppressive tumor microenvironment pose substantial challenges to effective treatment, necessitating the exploration of novel anticancer strategies. In contrast to synthetic drugs, natural compounds offer superior structural and biological diversity, often becoming a vast resource for anticancer drug discovery. In this investigation, we focused on ajuforrestin A, isolated from the plant *A. lupulina* Maxim, to assess its antitumor effects on A549 lung cancer cells.

Initially, we evaluated the in vitro inhibitory activity of ajuforrestin A using an MTT assay, which revealed a potent growth inhibition of A549 cells with an IC_50_ of 9.0 μM. Building on this finding, we explored the underlying anticancer mechanisms at the cellular level. It is established that inducing apoptosis and cell cycle arrest are two effective strategies for causing cancer cell death. Apoptosis, a principal form of programmed cell death, is a key therapeutic target [32]. Our experiments, using Annexin V-FITC/PI co-staining, showed a dose-dependent increase in apoptotic A549 cells upon treatment with ajuforrestin A. Since apoptosis is intricately linked to the cell cycle, we analyzed cell cycle progression. Flow cytometry revealed that ajuforrestin A induced a G0/G1 phase arrest in A549 cells, with the proportion of cells in this phase increasing significantly with the drug concentration.

The STAT3 transcription factor is frequently hyperactivated in NSCLC and is instrumental in promoting tumor-associated inflammation and immune evasion [23]. To our knowledge, the effect of ajuforrestin A on the STAT3 pathway in lung cancer has not been previously reported. Our Western blot analysis demonstrated that ajuforrestin A, while not affecting total STAT3 levels, significantly attenuated its activating phosphorylation. Apoptosis is a critical homeostatic process, and its induction is a primary goal of cancer therapy. This process is regulated by numerous proteins, including the Bcl-2 family, where Bcl-2 (anti-apoptotic) and Bax (pro-apoptotic) are key players [33]. Caspases, such as caspase-3 and caspase-9, are cysteine proteases that execute the apoptotic program. Our analysis showed that ajuforrestin A treatment led to the downregulation of Bcl-2 and the upregulation of Bax, alongside an increase in cleaved caspase-3 and caspase-9, confirming its pro-apoptotic mechanism via STAT3 pathway modulation.

FAK is intrinsically tied to the migration and invasion of tumor cells by mediating cytoskeletal reorganization and cell signaling [6]. This suggests FAK could be a general mediator of tumor metastasis and an early biomarker for invasion. MMP2, a downstream effector in the FAK pathway, is a common indicator of metastatic potential. Our research found that ajuforrestin A treatment significantly reduced the phosphorylation of FAK and the expression of MMP2 without altering total FAK levels. This indicates that ajuforrestin A can disrupt the FAK pathway to inhibit the invasion and migration of A549 cells, thereby impeding tumor progression.

Angiogenesis is also fundamental to tumor growth and malignant transformation, as it supplies tumors with necessary nutrients and oxygen [24]. Zebrafish models have become a standard tool for studying tumor angiogenesis and for drug development. To investigate the effect of ajuforrestin A on neovascularization, we used a transgenic zebrafish model. The results clearly showed that ajuforrestin A effectively disrupted new blood vessel formation. Pro-angiogenic factors, particularly VEGF, play a central role in this process by binding to receptors like VEGFR-2 on endothelial cells, activating signaling pathways that promote angiogenesis. We used SPR technology, a powerful tool for analyzing real-time biomolecular interactions, to examine the binding of ajuforrestin A to VEGFR-2. The data revealed a high-affinity interaction with a KD value of 45 μM.

Finally, to confirm the in vivo antitumor activity of ajuforrestin A, we utilized a zebrafish xenograft model with fluorescently labeled A549 cells. The results demonstrated that ajuforrestin A effectively suppressed both tumor growth and metastasis in a dose-dependent fashion within a living organism.

## 5. Conclusions

In this investigation, ajuforrestin A was identified to possess potent cytotoxic activity against A549 cells. Mechanistic studies showed that ajuforrestin A prompts apoptosis in A549 cells, with the apoptotic fraction increasing in proportion to the applied concentration. Flow cytometry experiments further established that ajuforrestin A induced a G0/G1 phase blockade in the cell cycle. Moreover, Western blotting analyses revealed that ajuforrestin A triggers apoptosis by modulating the STAT3 signaling pathway and curtails A549 cell migration via regulation of the FAK signaling cascade. As VEGFR-2 is a critical target in tumor angiogenesis, we demonstrated through zebrafish angiogenesis and SPR assays that ajuforrestin A can inhibit neovascularization by directly targeting this receptor. Ultimately, in vivo studies in a zebrafish model confirmed that ajuforrestin A significantly inhibited tumor proliferation and metastatic dissemination. In summary, the natural compound ajuforrestin A exerts considerable antitumor effects against A549 cells both in vitro and in vivo, presenting a novel therapeutic avenue for non-small cell lung cancer that merits further investigation.

## Figures and Tables

**Figure 1 biology-14-00908-f001:**
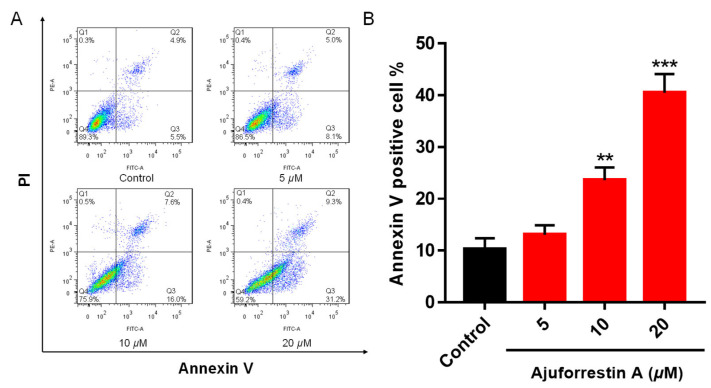
Apoptotic effects induced by ajuforrestin A in A549 cells. Cells were exposed to various concentrations of ajuforrestin A (5, 10, and 20 μM) for 48 h. Apoptosis was quantified using flow cytometry. (**A**) Representative flow cytometry plots for A549 cells under different treatments. (**B**) Bar graph showing the percentage of apoptotic cells after a 48-h exposure to ajuforrestin A. Data are shown as mean ± SD. ** *p* < 0.01 and *** *p* < 0.001 relative to the control group.

**Figure 2 biology-14-00908-f002:**
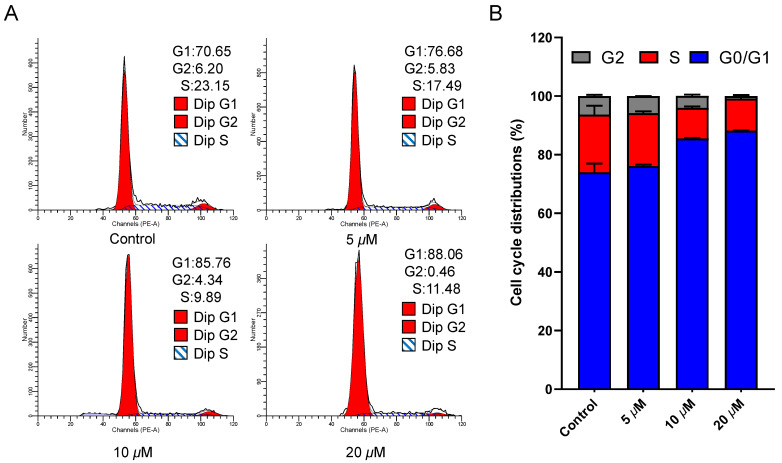
Impact of ajuforrestin A on the cell cycle distribution of A549 cells. Cells were incubated with escalating concentrations of ajuforrestin A (5, 10, and 20 µM) for 48 h and subsequently stained with propidium iodide (PI). (**A**) Cell cycle distributions were determined by flow cytometry. (**B**) The relative proportion of cells in different cell cycle phases following drug treatment. Data represent the mean ± SD from three independent experiments.

**Figure 3 biology-14-00908-f003:**
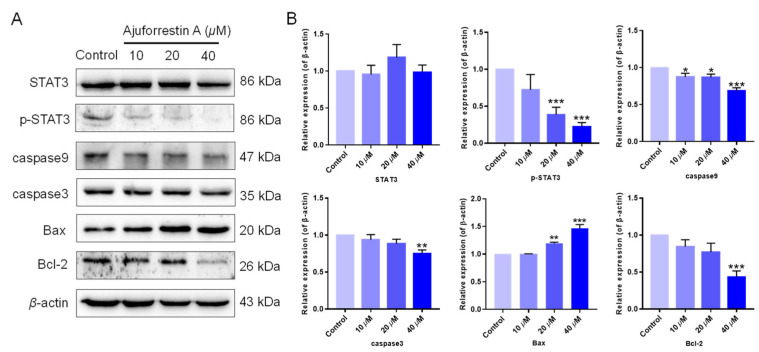
Influence of ajuforrestin A on the expression of STAT3-mediated apoptosis-related proteins in A549 cells. (**A**) A549 cells were treated with ajuforrestin A for 36 h. Western blotting was conducted to assess the expression of STAT3, p-STAT3, caspase-3, caspase-9, Bax, and Bcl-2. (**B**) Bar graph representing the relative protein expression levels compared to the control. Data are presented as mean ± SD. * *p* < 0.05, ** *p* < 0.01, and *** *p* < 0.001 relative to the control group.

**Figure 4 biology-14-00908-f004:**
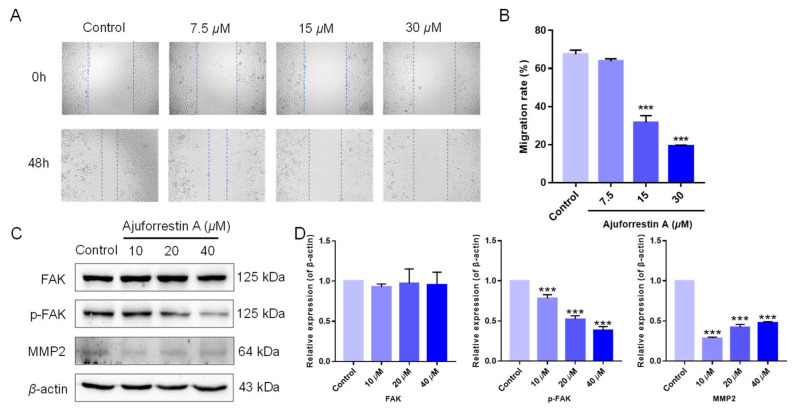
Ajuforrestin A impedes A549 cell migration through the FAK signal pathway. A549 cells were exposed to ajuforrestin A for 48 h, followed by Western blotting. (**A**) Representative micrographs of A549 cells at 0 and 48 h. (**B**) Bar graph of the migration rate. (**C**) Western blot results showing the expression of FAK, p-FAK, and MMP2. (**D**) Bar graph depicting the relative protein expression levels compared to the control. Data are expressed as mean ± SD. *** *p* < 0.001 relative to the control group.

**Figure 5 biology-14-00908-f005:**
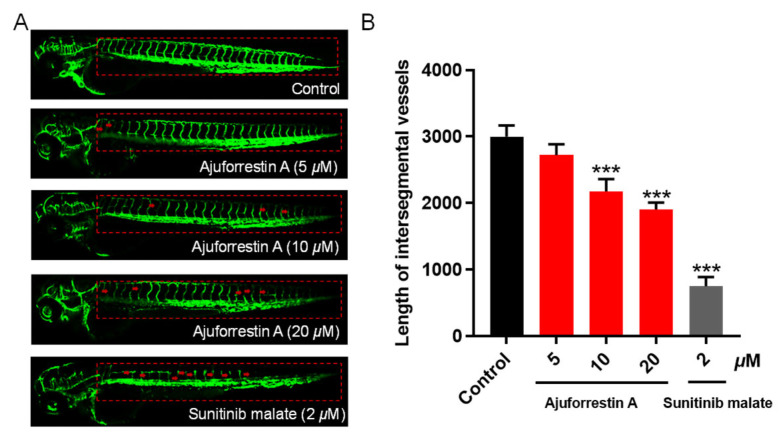
Anti-angiogenic activity of ajuforrestin A in a transgenic zebrafish model. Embryos from *Tg*(*fli1:EGFP*) zebrafish were exposed to ajuforrestin A (5, 10, and 20 μM) or the positive control, sunitinib malate (2 μM), for 48 h. (**A**) Representative confocal microscopy images of ISVs in zebrafish embryos. (**B**) The mean length of zebrafish ISVs after treatment with different concentrations of ajuforrestin A and sunitinib (n = 15/group). Data are presented as mean ± SD. *** *p* < 0.001 relative to the control group.

**Figure 6 biology-14-00908-f006:**
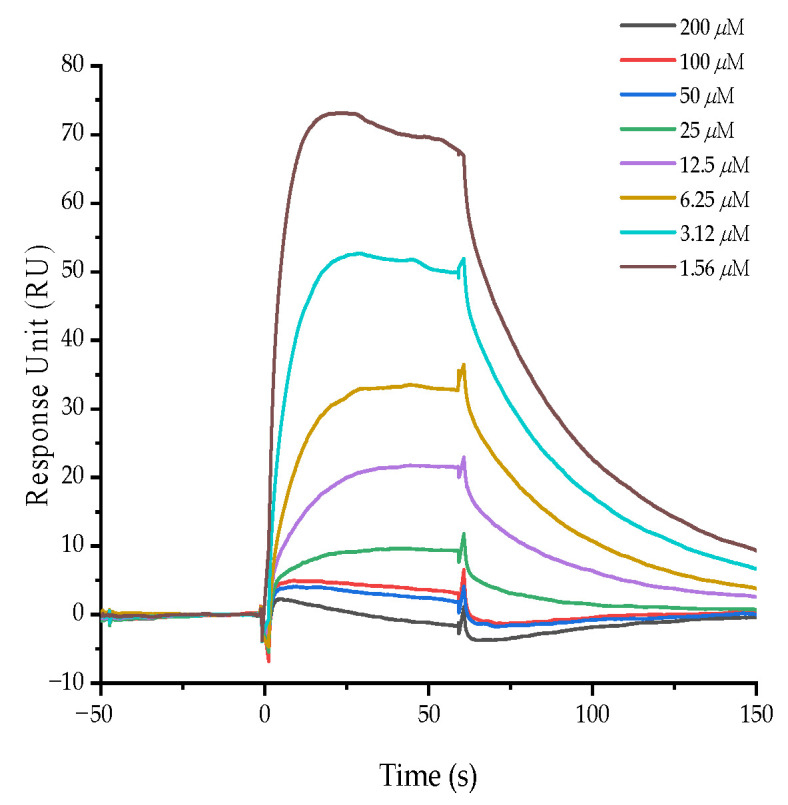
Affinity of ajuforrestin A for the VEGFR-2 protein as measured by SPR. The SPR sensorgram displays the binding response change (in RU) between varying concentrations of ajuforrestin A (1.56 to 200.0 μM) and immobilized VEGFR-2.

**Figure 7 biology-14-00908-f007:**
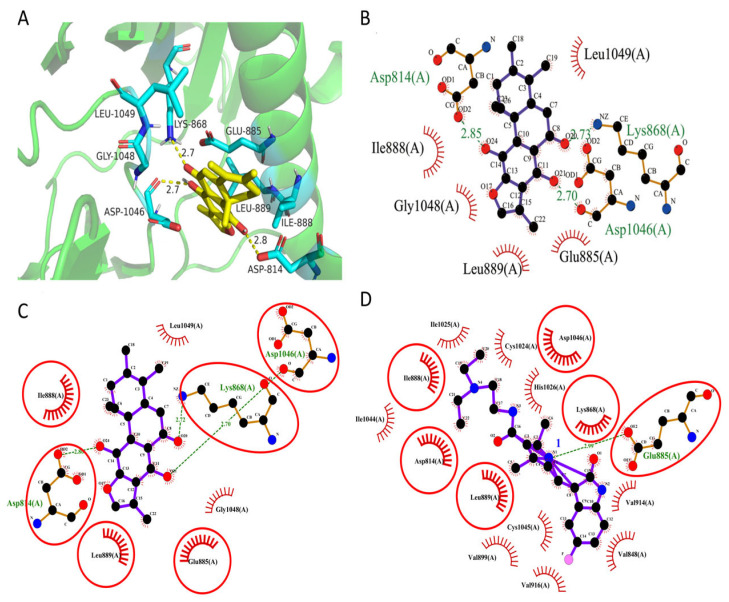
Predicted binding mode of ajuforrestin A within the VEGFR-2 active site. (**A**) A three-dimensional representation of the molecular interactions, showing hydrogen bonds and hydrophobic contacts of ajuforrestin A with the VEGFR-2 receptor protein (PyMOL). (**B**) A two-dimensional representation of the hydrogen bonds and hydrophobic interactions between ajuforrestin A and VEGFR-2 (LigPlot). (**C**,**D**) A side-by-side 2D interaction plot illustrating the interactions of Ajuforrestin A and sunitinib with the VEGFR-2 protein. The red circles represent the same residues that Ajuforrestin A and sunitinib interact with the VEGFR-2 protein.

**Figure 8 biology-14-00908-f008:**
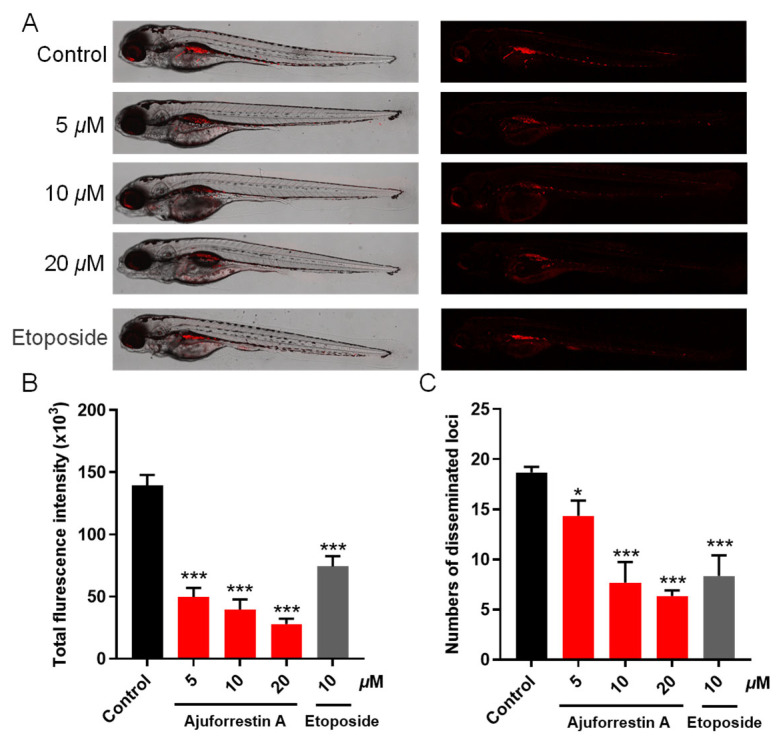
In vivo antitumor effects of ajuforrestin A in zebrafish xenografts. (**A**) Representative confocal microscope images showing the relative intensity and distribution of red fluorescence in zebrafish embryos. (**B**,**C**) Quantification of total fluorescence intensity and the number of disseminated foci from the tumor mass using ImageJ software (version 1.51k). Data are presented as mean ± SD. * *p* < 0.01, *** *p* < 0.001 relative to the control group.

**Table 1 biology-14-00908-t001:** Cytotoxicity of ajuforrestin A and etoposide against A549 lung cancer cell line.

Compounds	IC_50_ (μM) ^1^
Ajuforrestin A	9.0 ± 1.1
Etoposide	33.8 ± 1.0

^1^ The IC_50_ values were determined as described in the biological assays section. All results are presented as means ± SD.

## Data Availability

The original contributions presented in this study are included in the article/Appendix A. Further inquiries can be directed to the corresponding authors.

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
