# Peer review of "Ajuforrestin A Inhibits Tumor Proliferation and Migration by Targeting the STAT3/FAK Signaling Pathways and VEGFR-2"

_biology, 2025, doi:10.3390/biology14080908_

Round 1
Reviewer 1 Report
Comments and Suggestions for Authors
Experimental Focus and Significance
This study investigates the antitumor effects of ajuforrestin A, a natural diterpenoid isolated from Ajuga lupulina Maxim, on non-small cell lung cancer (NSCLC), specifically using the A549 lung cancer cell line. The researchers conducted a series of in vitro and in vivo experiments to evaluate the compound’s impact on cancer cell proliferation, apoptosis, cell cycle arrest, migration, and angiogenesis. Mechanistically, they found that ajuforrestin A exerts its effects primarily by modulating the STAT3 and FAK signaling pathways and by binding to VEGFR-2, thus inhibiting tumor angiogenesis.
The significance of this study lies in its identification of ajuforrestin A as a multi-targeted, low-toxicity natural compound with potential to serve as a lead candidate for NSCLC therapy. Its efficacy in both cellular and zebrafish models provides strong preliminary evidence supporting further preclinical development.
General Comments
To date, although several natural product-based studies have demonstrated anticancer effects on non-small cell lung cancer (NSCLC) by targeting individual pathways such as STAT3, FAK, or VEGFR-2, no prior research has specifically investigated ajuforrestin A in this context.
Research on the use of ajuforrestin A for NSCLC treatment has not been previously reported, and natural compounds that simultaneously target STAT3, FAK, and VEGFR-2 remain rare. Therefore, the multi-pathway inhibitory effects of ajuforrestin A represent a novel research area, highlighting the need for further exploration through preclinical and clinical studies to assess its therapeutic potential.
Major Concerns
- Lack of Mammalian In Vivo Validation
While the zebrafish xenograft model provides valuable insights, it does not fully replicate the complexity of the human tumor microenvironment. The absence of a mammalian in vivo model (e.g., mouse xenograft) limits the translational relevance of the findings and the assessment of pharmacokinetics or toxicity. - Insufficient Functional Validation of STAT3 and FAK Pathways
The study attributes the antitumor effects of ajuforrestin A to the inhibition of STAT3 and FAK signaling pathways based primarily on Western blot analysis. However, no pathway-specific knockdown (e.g., siRNA) or pharmacological rescue experiments were performed to establish a causal relationship. As a result, the mechanistic claims remain correlative rather than definitive. - SPR Binding Affinity and Functional Relevance
The authors report a KD of 45 μM for ajuforrestin A binding to VEGFR-2 based on SPR analysis. This relatively weak affinity raises questions about the biological significance of this interaction. Additional cellular assays (e.g., VEGFR-2 phosphorylation or downstream signaling inhibition in endothelial cells) would strengthen the claim that VEGFR-2 is a direct and relevant target. - Limited Justification of Dose Selection
The manuscript lacks sufficient rationale for the chosen concentrations of ajuforrestin A in both in vitro and in vivo experiments. It would be important to relate these doses to the IC₅₀ values and to assess whether observed effects could be due to non-specific cytotoxicity at higher concentrations. - Angiogenesis Assessment Limited to Zebrafish
The antiangiogenic activity of ajuforrestin A was evaluated solely using the Tg(fli1:EGFP) zebrafish model. While informative, the lack of complementary angiogenesis assays (e.g., HUVEC tube formation or aortic ring assays) weakens the generalizability of the antiangiogenic conclusion.
Author Response
Dear reviewer:
We would like to express our heartfelt gratitude for your meticulous review and valuable suggestions on our research work. Your comments have provided highly valuable guidance for further improving the manuscript, significantly enhancing the rigor and depth of the study. We have provided corresponding responses to your questions (the responses are included in the attachment).
Thank you again for your attention to the manuscript!Please check the attachement.
With kind regards!

Reviewer 2 Report
Comments and Suggestions for Authors
The manuscript entitled "Ajuforrestin A inhibits tumor proliferation and migration by targeting the STAT3/FAK signaling pathways and VEGFR-2" reports a comprehensive study on the antitumor potential of ajuforrestin A, a natural diterpenoid isolated from Ajuga lupulina Maxim. The authors investigate its effects on cell proliferation, apoptosis, migration, angiogenesis, and VEGFR-2 binding through a combination of in vitro, in vivo, and molecular docking approaches.
1 - Please include a validation step for your docking protocol, such as redocking the co-crystallized ligand of VEGFR-2 (from PDB ID: 4ASE) into its binding site. This would help confirm that the docking setup is reliable in reproducing the known pose.
2 - Please add the reference to the original publication associated with the 4ASE structure. This is standard practice to acknowledge the source of the structural data.
3 - There is no docking or comparative analysis involving a known VEGFR-2 inhibitor (positive control), such as the ligand co-crystallized in 4ASE or a clinically used VEGFR-2 inhibitor (e.g., sorafenib, sunitinib).
4- Please add a side-by-side 2D interaction plot comparing ajuforrestin A with the control ligand, highlighting shared and distinct critical residues.
5 - Please compare the interacting residues of ajuforrestin A with those of known inhibitors and discuss whether these residues are known to be important for VEGFR-2 activity.
6 - Please, if feasible, conducting molecular dynamics simulations for both the compound and the control ligand could significantly strengthen the robustness and impact of the findings.
Author Response
Dear Reviewer,
First and foremost, we sincerely thank you for your thorough review of our research and the valuable suggestions you have provided. Your comments have offered highly valuable guidance for improving the manuscript, particularly in the areas of molecular docking analysis, comparison with positive controls, and mechanism validation. These have significantly enhanced the rigor and depth of our study. We fully acknowledge the importance of your suggestion to supplement the comparative analysis with known inhibitors (such as sunitinib), which makes the mechanism of action of ajuforrestin A more convincing.
In accordance with your suggestions, we have completed the relevant experimental verifications, including docking analysis using AutodockTools, visual comparisons via Ligplot software, and detailed additions to the manuscript regarding interaction residues and binding energy comparisons (page 11). Furthermore, the 2D interaction diagram (Figure 7) you mentioned has been added as required, to visually demonstrate the binding pattern between the compound and the target. Your suggestion regarding molecular dynamics simulation also deeply reflects your concern for the completeness of the research. Although this study provides substantial evidence through SPR, molecular docking, and the zebrafish model, we fully recognize the importance of molecular dynamics simulations in dynamically validating the binding mechanism, and we plan to conduct related experiments in future research to further strengthen our conclusions. The responses are included in the attachment.
Once again, we sincerely thank you for your dedicated guidance and support of this research. Your suggestions have not only helped improve the current work but also provided important inspiration for future research directions. We believe that, driven by your professional insights, this study will be presented in a more rigorous manner.
